**Subject Category:**
Biology (whole organism)

evolution/palaeontology

placoderms, gnathostomes, neck joint, Silurian, *Silurolepis*

**Author for correspondence:**
Min Zhu
e-mail: zhumin@ivpp.ac.cn

# Reappraisal of the Silurian placoderm *Silurolepis* and insights into the dermal neck joint evolution

You-an Zhu[1,2,3], Jing Lu[1,2] and Min Zhu[1,2,4]

[1]Key Laboratory of Vertebrate Evolution and Human Origins, Institute of Vertebrate Paleontology and Paleoanthropology, Chinese Academy of Sciences (CAS), 142 Xi-zhi-men-wai Street, Beijing 100044, People's Republic of China
[2]CAS Center for Excellence in Life and Paleoenvironment, Beijing 100044, People's Republic of China
[3]Subdepartment of Evolution and Development, Uppsala University, Norbyvägen 18A, 752 36 Uppsala, Sweden
[4]College of Earth and Planetary Sciences, University of Chinese Academy of Sciences, Beijing 100049, People's Republic of China

Y-aZ, 0000-0002-6911-540X; JL, 0000-0002-5791-4749; MZ, 0000-0002-4786-0898

*Silurolepis platydorsalis*, a Silurian jawed vertebrate originally identified as an antiarch, is here redescribed as a maxillate placoderm close to *Qilinyu* and is anteroposteriorly reversed as opposed to the original description. The cuboid trunk shield possesses three longitudinal cristae, obstanic grooves on the trunk shield and three median dorsal plates, all uniquely shared with *Qilinyu*. Further preparation reveals the morphology of the dermal neck joint, with slot-shaped articular fossae on the trunk shield similar to *Qilinyu* and antiarchs. However, new tomographic data reveal that *Qilinyu* uniquely bears a dual articulation between the skull roof and trunk shield, which does not fit into the traditional 'ginglymoid' and 'reverse ginglymoid' categories. An extended comparison in early jawed vertebrates confirms that a sliding-type dermal neck joint is widely distributed and other types are elaborated in different lineages by developing various laminae. Nine new characters related to the dermal neck joint are proposed for a new phylogenetic analysis, in which *Silurolepis* forms a clade with *Qilinyu*. The current phylogenetic framework conflicts with the parsimonious evolution of dermal neck joints in suggesting that the shared trunk shield characters between antiarchs and *Qilinyu* are independently acquired, and the sliding-type joint in *Entelognathus* is reversely evolved from the dual articulation in *Qilinyu*.

# 1. Introduction

Silurian gnathostome fossils used to be scarce and mostly disarticulated [1,2], despite the fact that morphologically distinctive major groups of Devonian jawed vertebrates suggest a long and sophisticated history in Silurian [3]. Discovered in the Ludlow Kuanti Formation of Qujing, Yunnan, China, and first mentioned in 1993 [4], *Silurolepis platydorsalis* was one of the most complete Silurian jawed vertebrates. The extensively developed macromeric dermal plates show distinctive patterns comparable to the Devonian placoderms, suggesting that *Silurolepis* is also a placoderm. Specifically, *Silurolepis* was first identified as an antiarch [5], based on the cuboid trunk shield with the putative two median dorsal (MD) plates, which were then considered to be exclusive in antiarchs [2].

A series of recent discoveries in the Silurian Xiaoxiang Fauna [6–10] from a neighbouring site of the Kuanti Formation reveal a completely new group of placoderm-grade jawed vertebrates, collectively known as maxillate placoderms [8,11], with exquisite preservation of articulated skeletons. The first one, *Entelognathus primordialis*, displays an unexpected character combination with skull roof and trunk shield resembling a typical placoderm, and osteichthyan-like maxillate and premaxillate jaw bones [6]. The second maxillate placoderm discovered, *Qilinyu rostrata*, combines the marginal jaw bones with characters from different placoderm subgroups [8]. Notably, the cuboid trunk shield with multiple MD plates in *Qilinyu* resembles the condition in antiarchs. These new discoveries significantly complicated the long-standing debates over the placoderm systematics, the inter-relationships of placoderm subgroups, and the earliest evolution of jawed vertebrates.

Placoderms are traditionally classified into well-established subgroups, with arthrodires and antiarchs being the two most diverse and representative clades [12]. They are consistently regarded as a representative of the earliest jawed vertebrate morphotype, regardless of their debatable phylogenetic assignments [13,14]. Seemingly inconsistent with the primitive position, most placoderm subgroups are known to emerge in the Early Devonian radiation of jawed vertebrates, coeval with bony and cartilaginous fishes [15,16]. Consequently, there is still controversy regarding the polarities of some, often key characters of the Devonian placoderms [17–19]. The recently discovered Silurian placoderms, including *Silurolepis*, are chronologically closer to the earliest radiation of jawed vertebrates and show character combinations that are distinctive to the later placoderms. Detailed knowledge of these Silurian taxa, compared with other traditionally defined placoderms, is expected to yield further data that are key in assessing primitive gnathostome characters.

New evidence from *Entelognathus* and *Qilinyu*, especially the latter with cuboid trunk armour and multiple MD plates, which were previously only known in antiarchs, also severely undermines the original assignment of *S. platydorsalis* as an antiarch. In this paper, we provide a revised description of *S. platydorsalis* based on both further preparation of the holotype, and an updated comparison with other placoderms including the maxillate placoderms. Based on the evidence, we refute the previous identification of *S. platydorsalis* as an antiarch and suggest that it is in fact a maxillate placoderm close to *Qilinyu*.

# 2. Material and methods

## 2.1. Materials

The type specimen of *Silurolepis* (IVPP V11680.1) was further mechanically prepared from the matrix to reveal the anatomical details. The type specimen of *Qilinyu* (IVPP V20732) and an additional articulated specimen of *Qilinyu* (IVPP V20733.1) were investigated. All specimens are housed in the Institute of Vertebrate Paleontology and Paleoanthropology (IVPP), Chinese Academy of Sciences (CAS), Beijing, China.

## 2.2. X-ray tomography

The type specimen of *Qilinyu* (IVPP V20732) was scanned at IVPP, using 225 kV micro-CT (developed by the Institute of High Energy Physics, CAS), with a voltage of 150 kV and current of 120 mA, at a resolution of 41.56 µm per pixel. The scan was conducted using a 720° rotation with a step size of 0.5° and an unfiltered aluminium reflection target. A total of 720 transmission images were reconstructed in a 2048 × 2048 matrix of 1536 slices. Three-dimensional segmentation was performed in Mimics 18.0 (https://www.materialise.com/en/medical/software/mimics; Materialize). Surface meshes were then exported into, dislodged, surface rendered and imaged in Blender 2.79b (http://blender.org; Stitching

Blender Foundation, Amsterdam, The Netherlands). The images of the reconstructions were finalized in Adobe Photoshop and Adobe Illustrator.

## 2.3. Phylogenetic analysis

To explore the phylogenetic position of *Silurolepis* and the impact of the new neck joint characters on gnathostome phylogeny, we conducted a phylogenetic analysis based on the dataset consisting of 379 characters and 105 taxa in total, which was sourced from Zhu *et al.* [8] by replacing two characters related to the dermal neck joint with nine new characters. The character data entry and formatting were performed in Mesquite (v. 2.5). All the characters were unordered.

The dataset was subjected to maximum-parsimony analysis in the TNT software package. Galeaspid was set to be the outgroup. The analysis was conducted using a new technology search strategy, with 100 000 maximum trees in memory and 1000 random additional sequences. Bremer support values were generated in TNT using the script 'aquickie.run'. The analysis produced 22 most parsimonious trees of 1061 steps each. Consistency index (CI) = 0.379; retention index = 0.808.

# 3. Results

## 3.1. Systematic palaeontology

Gnathostomata Gegenbaur, 1874 [20]

*nomen nudem* 1993 *Silurolepis platydorsalis*—Wang, 1993 [4]
*nomen nudem* 2000 *Silurolepis platydorsalis*—Zhu & Wang, 2000 [21]
*Silurolepis platydorsalis*—Zhang *et al.*, 2010 [5]

*Revised diagnosis*: A moderately large maxillate placoderm with cuboid trunk shield. Key characters include the anterior margin of the trunk shield wider than the posterior; three longitudinal cristae along the median and lateral edges of the cuboid trunk shield; main lateral line immediately ventral to the lateral cristae; a small first, a large second and a possible third MD plate; anterior dorsolateral (ADL) plate significantly shorter than the posterior dorsolateral plate; dermal neck joint similar to *Qilinyu* and antiarchs, with a slot-shaped articular fossa along the anterodorsal margin of the trunk shield, presumably for the insertion of the posterior margin of the skull roof; and an obstanic groove along the anterolateral margin of the trunk shield putatively receiving the posterolateral margin of the skull roof.

*Horizon and locality*: The holotype (IVPP V11680.1) and the referred specimen (IVPP V11680.2) of *S. platydorsalis* were collected by Shi-tao Wang and Nian-zhong Wang in 1970s and 1980s, respectively. The specimens were from a Silurian site near the dam of the Xiaoxiang Reservoir, in the suburb of Qujing City, Yunnan Province, China [5]. The site, although close by, is different to the one that yields *Guiyu*, *Entelognathus* and *Qilinyu*, and the two may belong to different horizons of the Kuanti Formation. The exact sequence correlation between these two sites needs further investigation.

## 3.2. Description

At the time when *Silurolepis* was first described, the Antiarcha was the only placoderm group that possessed a cuboid trunk shield incorporating multiple MD plates, and distinctive paired dorsolateral ridges. As such, *Siluriolepis* was described with a reference to a generalized antiarch pattern [5]. Shared characters include the anterior MD plate larger than the posterior one, and anteriorly tapered trunk shield. Several unusual conditions, including the anterior MD plate overlapping the posterior one, and a broad anterior margin of the anterior MD plate, were interpreted as derived within antiarchs. Recent discovery of a new Silurian placoderm *Qilinyu* in an adjacent locality, with complete exoskeleton preservation, provides convincing evidence for a new interpretation of the shield orientation in *Silurolepis*. *Qilinyu* possesses three MD plates, with the anterior two plates matching the two MD plates of *Silurolepis* in proportion, outline and overlapping relationship. The large ADL plate and elongated cuboid trunk shield also coincide with the condition in *Silurolepis*. The third MD plate in *Silurolepis* is possibly missing due to the incomplete preservation. Notably, in *Qilinyu*, the third MD is easily detached from the rest of the trunk armour, and a number of specimens are preserved without the third MD plate. Based on this new insight, we conclude that the shield orientation in the original description of *Siluriolepis* should be reversed and the holotype

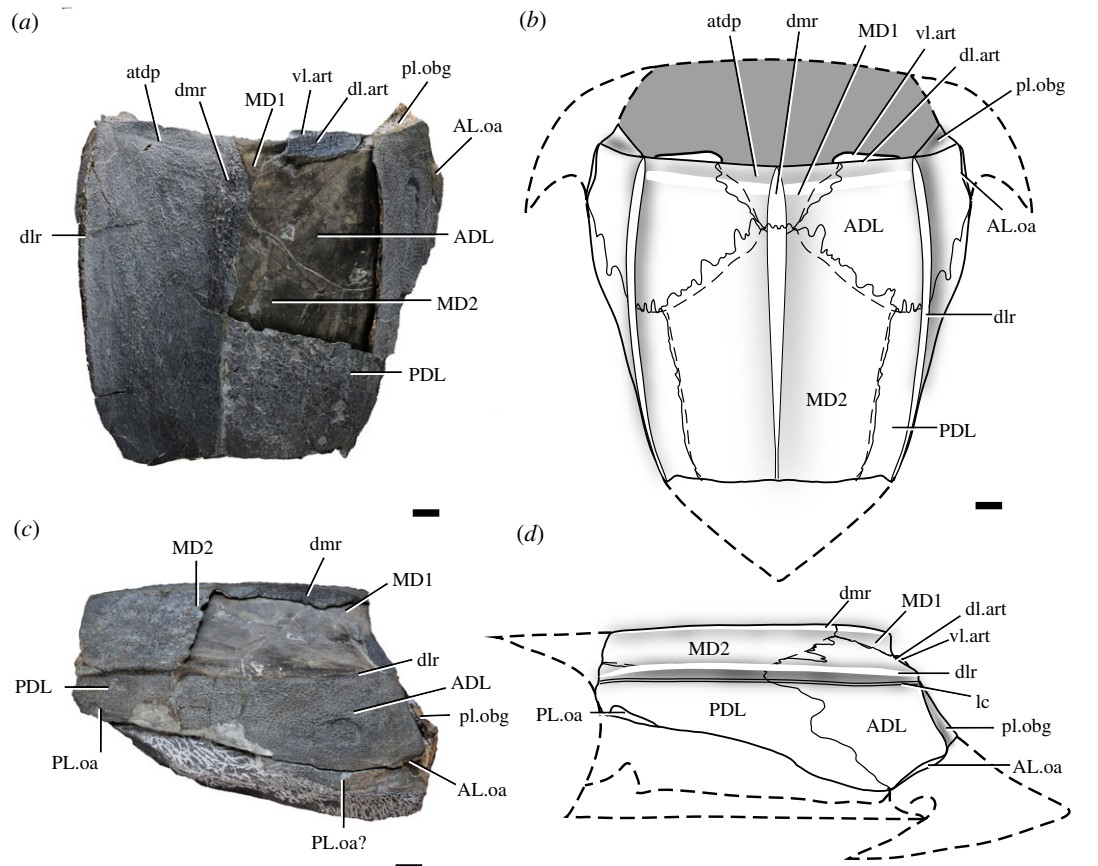

**Figure 1.** *Silurolepis platydorsalis* (holotype, IVPP V11680.1). (*a*) In dorsal view. (*b*) Reconstruction of the trunk shield in dorsal view. (*c*) In lateral view. (*d*) Reconstruction of the trunk shied in lateral view. Scale bar = 1 cm. Abbreviations: ADL, anterior dorsolateral plate; AL.oa, overlap area for the anterolateral plate; atdp, anterior transverse depression; dlr, dorsolateral ridge of the trunk shield; dl.art, dorsal lamina of the articular fossa on the trunk shield; dmr, dorsomedian ridge of the trunk shield; lc, main lateral line canal; MD1, first median dorsal plate; MD2, second median dorsal plate; PDL, posterior dorsolateral plate; PL.oa, overlap area for the posterolateral plate; pl.obg, posterior lamina of the obstanic groove; vl.art, ventral lamina of the articular fossa on the trunk shield.

represents a maxillate placoderm close to *Qilinyu*. The reorientation is confirmed by the discovery of the neck joint structures in the holotype after the further preparation. An updated description is provided below.

The trunk shield of *Silurolepis* is preserved only with its dorsal and lateral aspects. The preserved parts suggest roughly a cuboid shape. The dorsal wall is almost flat with only slight arching along the median line, unlike the strongly arched trunk shield in *Qilinyu*. The dorsal wall meets the lateral wall in an angle of approximately 130° at the anterior margin and the near-right angle at the posterior preserved margin (figures 1 and 2). This new reconstruction renders the whole trunk shield proportionately wider and shorter than the previous reconstruction [5]. Three longitudinal cristae or ridges run through the dorsal surface of the trunk shield: the median ridge (dmr, figures 1 and 2) and the paired dorsolateral ridges (dlr, figures 1 and 2). The dermal tubercular ornament is dense along these cristae and is densest around the anterior part of the median ridge. Similar pronounced longitudinal ridges are also present in most antiarchs [22–24], but the homology of the ridges is unclear under the current phylogenetic framework. All these ridges in *Silurolepis* are water-drop-shaped, broadening anteriorly and tapering posteriorly. The main lateral line canal (lc, figure 1*c*,*d*) runs along and immediately ventral to the dorsolateral ridge. Dorsally, a shallow groove traverses the most anterodorsal margin of the trunk shield (atdp, figure 1*a*,*b*); on the visceral surface, a corresponding transverse thickening is present (th.tr, figure 2*b*). Additionally, a stripe of visceral thickening is present along the dorsolateral edge of the posterior dorsolateral plate (PDL) and significantly widened at the posterior end (th.pl, figure 2*b*). The transverse groove and visceral thickenings are not present in *Qilinyu*.

The sutures between individual plates are vaguely traceable on the external surface and highly sinuous (figure 1). The sutures also can be recognized on the internal surface and on the mould (figure 2*a*). The internal sutures do not display the sinuous pattern as in the external ones. In the original description,

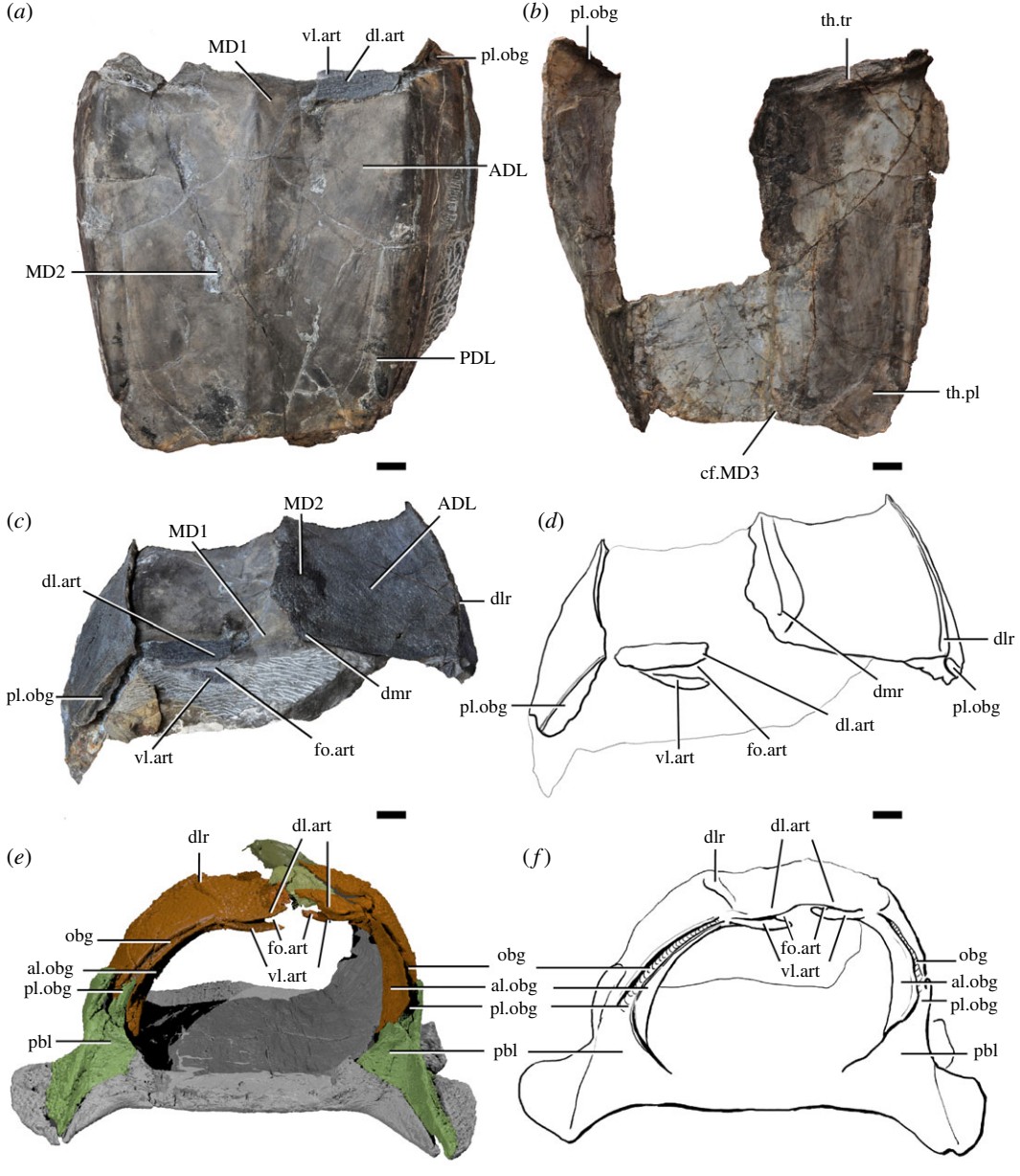

**Figure 2.** Trunk shields of *Silurolepis platydorsalis* and *Qilinyu rostrate*. (*a*) Internal mould of the trunk shield of *Silurolepis platydorsalis* (holotype, IVPP V11680.1), in dorsal view. (*b*) Trunk shield of *Silurolepis platydorsalis* in ventral view, showing the visceral morphology. (*c*) Trunk shield of *Silurolepis platydorsalis* in anterior view. (*d*) Interpretative drawing of the trunk shield of *Silurolepis platydorsalis* in anterior view. (*e*) Digital reconstruction and (*f*) interpretive drawing of the trunk shield of the *Qilinyu rostrate* holotype (IVPP V20732) in anterior view. Scale bar = 1 cm. Abbreviations: ADL, anterior dorsolateral plate; al.obg, anterior lamina of the obstanic groove; cf. MD3, contact face for the third median dorsal plate; dl.art, dorsal lamina of the articular fossa on the trunk shield; dlr, dorsolateral ridge of the trunk shield; dmr, dorsomedian ridge of the trunk shield; fo.art, articular fossa on the trunk shield; MD1, first median dorsal plate; MD2, second median dorsal plate; obg, obstanic groove receiving the posterolateral margin of the skull roof; pbl, posterior branchial lamina; PDL, posterior dorsolateral plate; pl.obg, posterior lamina of the obstanic groove; th.tr, anterior transverse thickening; th.pl, posterolateral thickening; vl.art, ventral lamina of the articular fossa on the trunk shield.

*Silurolepis* was interpreted as having large anterior and small posterior MD plates [5]. Under our new interpretation, the previous posterior MD plate is the first MD plate (MD1, figures 1 and 2), which is triangular, and the previous anterior MD plate is in fact the second MD plate (MD2, figures 1 and 2). The contour and proportion of both these two MD plates resemble that in *Qilinyu*. Both the two MD plates do not bear the visceral keel. Judging from the extension and contact face (cf.MD3, figure 2*b*) on the posterior edge of the second MD plate, there is a third MD plate as in *Qilinyu*.

Two pairs of large plates, namely the ADL and the PDL plates, flank the MD plates and bear the paired dorsolateral ridges and the main lateral lines (figures 1 and 2). A similar pattern is again present in *Qilinyu*. In both taxa, the ADL plate is roughly semicircular, and the PDL is a slender trapezoid plate. However, in *Qilinyu*, the ADL is significantly larger and longer than the PDL. By contrast, the ADL in *Silurolepis* is significantly shorter in longitudinal length than the PDL, although the dorsal lamina of the former is still larger than that of the latter. The anteroventral corner of the ADL plate bears an overlap area (AL.oa, figure 1), evidently for the unpreserved anterior lateral (AL) plate. Notably, the short edge for the overlap area denotes a small-sized AL plate, in comparison with the much larger ADL plate. The ADL larger than the AL is also present in *Qilinyu*. By contrast, in most other placoderms including *Entelognathus*, acanthothoracids, ptyctodonts, petalichthyids and basal arthrodires such as *Sigaspis* [25], the ADL is much reduced and nearly always significantly smaller than the AL. It is worth noting that in primitive osteichthyans, such as *Guiyu* and *Psarolepis*, the dorsal segment of the shoulder girdle, presumably equivalent to the ADL plate in placoderms, is underdeveloped, while the lateral lamina of the cleithrum, presumably homologous to the AL plate, is proportionately comparable to the equivalent plate in most placoderms but not in *Silurolepis*, *Qilinyu* and antiarchs [7,26].

Two significant structures related to the head–trunk interface are present on the trunk shield of *Silurolepis*. Both structures are also shared in *Qilinyu*. Firstly, a lamina is present along the right anterolateral margin of the trunk shield (pl.obg, figures 1 and 2*a*–*d*). The dorsal part of the same lamina is also present on the left side. In the trunk shield of *Qilinyu*, which is completely preserved, a similar lamina (pl.obg, figures 2*e*,*f* and 3*a*,*b*,*d*), is also present, with a prominent extension curled laterally to develop an anterior lamina (al.obg, figures 2*e*,*f* and 3*a*,*b*,*d*). The two laminae together form a U-shaped groove opened laterally (obg, figures 2*e*,*f* and 3*a*,*b*,*d*). The articulated skeletons in multiple specimens of *Qilinyu* allow a confident reconstruction of the contact relationship between the skull roof and trunk shield (figure 3). In *Qilinyu*, this groove is evidently an obstanic structure that receives the posterolateral margins of the skull roof (obt.sr, figure 3*a*,*c*), when the head shield is depressed around the neck joint. The posterolateral margin of the head shield is curled inward to be fitted into the laterally opened groove. The posterior lamina of the groove, not the anterior lamina, is continuous to the ventrally positioned postbranchial lamina (pbl, figures 2*e*,*f* and 3*a*,*c*). The anterior lamina is not preserved in *Silurolepis*. However, the broken mesial edge of the posterior lamina just starting to curl and the dorsal position of this lamina on the ADL, which is not seen in other placoderms, imply a similar groove and pattern of contact between the dermal head and trunk skeleton to *Qilinyu*.

Secondly, a ventral lamina under the anterodorsal margin of the trunk shield, which was newly revealed by further preparation, is recognized as the ventral lamina for the neck joint (vl.art, figures 1 and 2*a*–*d*). Correspondingly, the flange dorsal to this lamina continuous with the rest of the anterodorsal margin of the trunk shield, is the dorsal lamina of the neck joint (dl.art, figures 1 and 2*a*, *c*,*d*), and the slot-like concavity between these two laminae is the articular fossa (fo.art, figure 2*c*,*d*). The ventral lamina is transversely elongated, and is quite thin, in contrast to the thickened articular lamina or condyles in some other placoderms such as brachythoracid arthrodires and ptyctodonts [27–29]. Detailed comparison of different types of the dermal neck joint in placoderms and their implication are provided below.

## 4. Discussion

### 4.1. Evolution of dermal neck joint in jawed vertebrates

The head–trunk boundary, on which a neck is defined and formed, is considered as one of the fundamental interfaces in the evolution and development of the vertebrate body [30–34]. Despite the well-defined differentiation of the endoskeleton between head and trunk in all craniates, this boundary is primitively not presented in the dermal skeleton in the total-group gnathostomes. In most ostracoderms or jawless stem gnathostomes, including galeaspids and osteostracans which are closely related to jawed vertebrates, the dermal skeleton that covers the anterior half of the body is one integrated box of armour (figure 4). A dermal head–trunk boundary only appeared after the jawless–jawed transition and is to date still one synapomorphy exclusively shared in jawed vertebrates. Notably, the contact between dermal head and trunk shield forms a distinct articulation in nearly all placoderm-grade jawed vertebrates. In early bony fishes, although the dermatocranium and dermal shoulder girdle are still closely associated, the articulation becomes an indistinct overlap and movement is restricted. Further, the shoulder girdle is completely separated from the cranium, independently in some coelacanths [35] and in two

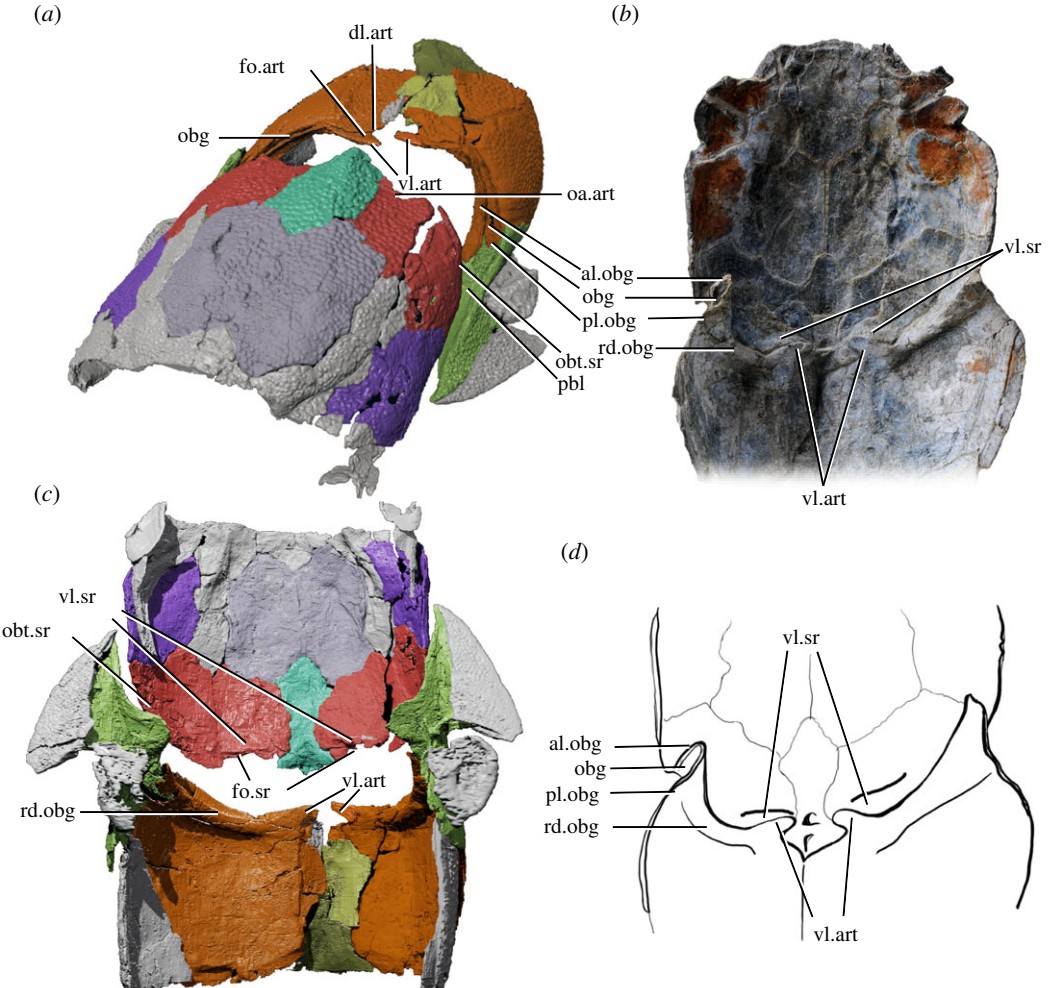

**Figure 3.** *Qilinyu rostrate* (IVPP V20732). (*a*) Digital reconstruction and rendering of the holotype (IVPP V20732) based on new tomographic data, in anterior left lateral view, with the cephalic and trunk shield artificially dislocated a bit from the original position, to reveal the dermal cervical articulation. (*b*) IVPP V20733.1 in ventral view, showing the articulated condition. Only skull roof and the dorsal part of the trunk shield are preserved in this specimen. (*c*) Digital reconstruction of the holotype in ventral view, with the ventral aspect removed, and cephalic and trunk shield dislocated. (*d*) Interpretative drawing of IVPP V20733.1 in ventral view. Not to scale. Abbreviations: al.obg, anterior lamina of the obstanic groove; dl.art, dorsal lamina of the articular fossa on the trunk shield; fo.art, articular fossa on the trunk shield; fo.sr, articular fossa on the skull roof; oa.art, overlap area on the skull roof, overlapped by the dorsal lamina of the articular fossa on the trunk shield; obg, obstanic groove receiving the posterolateral margin of the skull roof; obt.sr, posterolateral margin of the skull roof received by the obstanic groove; pbl, posterior branchial lamina; pl.obg, posterior lamina of the obstanic groove; rd.obg, visceral ridge corresponding to the obstanic groove; vl.art, ventral lamina of the articular fossa on the trunk shield; vl.sr, ventral lamina of the articular fossa on the skull roof.

tetrapodomorph lineages [36,37], preventing the formation of any form of dermal neck joint. In the latter group, this separation paves the way for a highly kinetic neck adapting to the terrestrial lifestyle.

The different types of dermal neck joint in primitive jawed vertebrates have been discussed by various authors [27,38–42]. It was generally accepted that a sliding joint is primitive in placoderms and the structurally specialized joints, including ginglymoid and reverse ginglymoid, are independently evolved in different lineages. However, it was not specified how the complicated articulations and accessory para-articular structures arose from the primitive sliding-type joint. Accordingly, in recent phylogenetic analyses of early gnathostomes, only generalized characters such as presence or absence of dermal neck joint, and joint types being 'sliding', 'ginglymoid', 'reverse ginglymoid' and 'spoon-like' are defined. Under the generalized classification, the possible transitions and intermediates between primitive sliding joint and different types of complicated joint are not illustrated, and their phylogenetic implication is not fully assessed.

The discoveries of the Silurian placoderms *Qilinyu* and *Silurolepis* further complicate the traditional categories of dermal neck joint in placoderms. Although the neck joint of *Qilinyu* was not discussed in

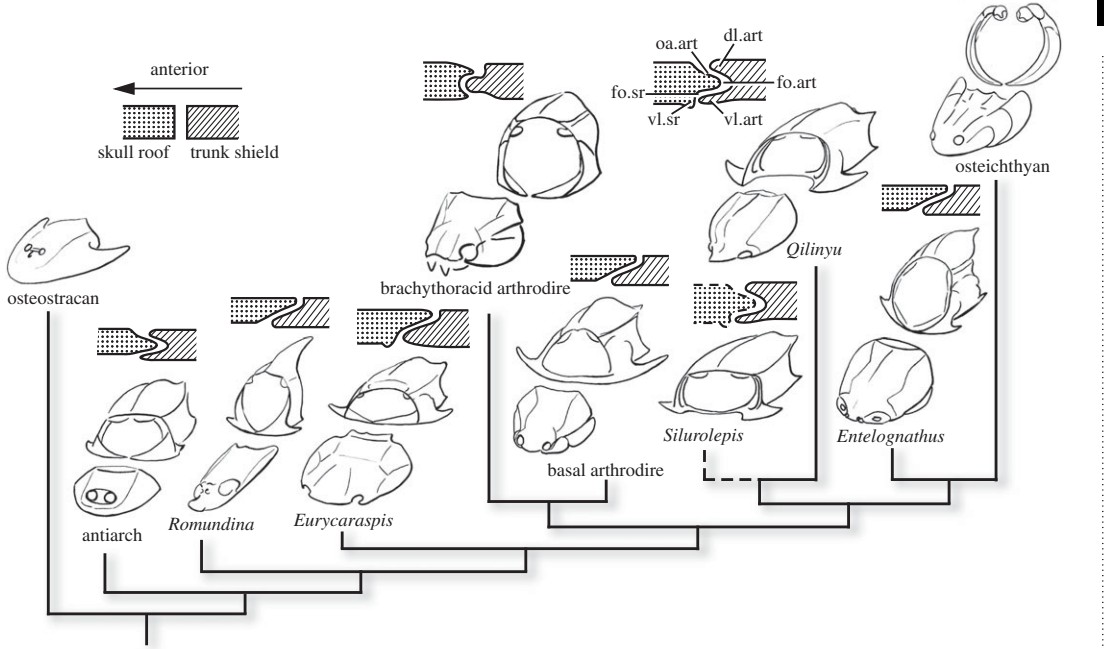

**Figure 4.** Evolution of dermal neck joint in the earliest jawed vertebrates, detailing on the contact relationship at the articular point (where the main lateral line crosses the head–trunk boundary). The phylogeny is a summary from [8].

the original description [8], additional specimens and new tomographic data now allow a detailed investigation. Evidently, the articular structures on the trunk shield are very close to that of *Silurolepis*, showing a pair of long, slot-like articular fossae (fo.art, figure 3*a,c*) almost meeting at the midline. The skull roof further reveals a new type of articulation unknown in any other placoderms. The posterior margin of the skull roof is interposed into the slot-shaped articular fossa on the trunk shield resembling that in antiarchs. However, the protruding ventral lamina on the trunk shield in turn inserts under a second, shallow and ventrally open fossa (fo.sr, figure 3*a,c*) on the posterior paranuchal plate of the skull roof, anteriorly confined by a ridge-like lamina (vl.sr, figure 3*b–d*). The movement of this dual articulation must be fairly limited, as the posterior end of the skull roof, when interposed into the articular fossa, is overlapped and constricted by the first MDl plate and the dorsal lamina of the articulation on the ADL plate (figure 3*b,d*). However, because the articulation is very close to the midline, effectively making a very short side of a lever, the whole head is still allowed to make considerable pitch movement, acting as the long side of the lever. When the head is depressed, the inwardly curled posterolateral margin of the skull roof (obt.sr, figure 3*a,c*) is apparently accommodated by the laterally opened groove on the ADL and AL plates (obg, figure 3*a,b,d*) between the anterior and the posterior laminae of the groove.

The dual articulation in *Qilinyu* and possibly in *Silurolepis* cannot be accommodated by any of the traditional categories of neck joint types. As such, here we focus on tracking the distribution and evolution of component structures forming the dermal skeleton contact between head and trunk, in various early gnathostome groups. Also, we make a clear distinction between the direct articulating region of the joint, from various para-articular structures which were often included to define a collective type of joint, but are easily demonstrated to be independent characters.

As noted by previous studies, the sliding type of joint, consisting of a trunk element (ADL plate) overlapped by the skull roof element (paranuchal) and allowing movement to a certain extent, is primitively present across a wide range of placoderm subgroups, including acanthothoracid such as *Romundina* [43], basal arthrodires such as *Kujdanowiaspis* [44], and the recently discovered maxillate placoderm *Entelognathus* [6]. *Brindabellaspis* is unique in having the cranial articulation for the 'dermal' neck joint carried by the endoskeletal braincase, rather than the dermal skull roof [45]. The overlapped area on the trunk shield can be developed into differentially shaped flanges or laminae in various placoderm groups. The thickened and narrow flanges are sometimes termed 'condyle' and often bear rotatory contacting surfaces as part of the definition of ginglymoid articulations, such as those in brachythoracid arthrodires. We tentatively termed this flange the articular lamina. Evidently, the articular lamina, or the homologous structure known as flange or condyle, is present in virtually all types of the neck joints and forms the basic component of the dermal neck joint in placoderms.

The neck joint in antiarchs is traditionally considered unique in placoderms and is dubbed 'reverse ginglymoid', in which the posterior margin of the skull roof is extended as an articulating flange into a horizontal fossa on the anterodorsal part of trunk shield. Essentially, nearly identical conditions occur in *Qilinyu* and *Silurolepis*, in which the posterior flange of the skull roof inserts into the slot-shaped articular fossa on the trunk shield. The paired ventral articular lamina in antiarchs, named 'subarticular ridge' [46], forms the ventral aspect of the slot-shaped fossa and should be homologous to the articular lamina in *Qilinyu*, *Silurolepis* and most other placoderms. Although in antiarchs the ventral surface of the skull roof does not bear a fossa as in the dual articulation of *Qilinyu*, the articular fossa on the trunk shield, a continuous flange on the posterior margin of the skull roof acting like articulation lamina and the cuboid trunk shield incorporating multiple MD plates, are still only shared in antiarchs, *Qilinyu* and *Silurolepis*. The deep obstanic groove along the anterolateral margin of the trunk shield seems to be exclusively shared by *Qilinyu* and *Silurolepis*. An obstanic lamina is present on the trunk shield of some arthrodires such as *Dicksonosteus* [47], but in those cases, the obstanic lamina is flat and does not develop into a prominent groove with a curled anterior lamina, as in *Qilinyu* and *Silurolepis*. Note that the posterior wall or lamina, not the anterior lamina of the obstanic groove, is continuous with the postbranchial lamina carried mainly by the AL plate (pbl, figures 2*e,f* and 3*a*).

The dermal neck joint in the petalichthyid *Eurycaraspis* displays a condition closer to an elaborated sliding-type joint. The trunk shield of *Eurycaraspis* bears a simple, moderately thickened articular lamina inserted under the skull roof, like that in a typical sliding-type joint. The contact face on the skull roof for the articular lamina, however, is restricted mesially by the paired lateral process [48], in addition to a median posterior descending lamina overlapped by the extrascapular plate. The same lateral process is also present in the putatively basal petalichthyid *Diandongpetalichthys* [49], which lacks a posterior descending lamina. The ptyctodont *Rhamphodopsis* possesses similar lateral process [39] mesial to the articulation fossa, and an articular lamina on the ADL plate. However, other ptyctodonts lack the lateral process and develop a very robust articular condyle, which differs from the condition in brachythoracid arthrodires in having concave or 'spoon-like', rather than convex and rotatory, facets [27,50–52]. Stensiö interpreted that, in the ptyctodontid *Chelyophorus*, the condyle on the ADL plate actually develops a fossa-like facet for a 'double joint' [53]. In some brachythoracid arthrodires such as the enigmatic heterosteids with highly kinetic neck joints, the lateral corner of the paranuchal contacts a long and shallow concavity on the ADL plate dorsal to the articular condyle, presumably to support and stabilize the joint [54]. In both cases, they are distinct from the dual articulation in *Qilinyu*.

## 4.2. Proposed characters related to the dermal neck joint for future phylogenetic analyses

The traditionally defined categories of neck joint types in placoderms obviously represent a composite of characters, and the transitions between these types can be achieved by the stepwise development or reduction of independent structures. For example, the 'ginglymoid joint' can be formed from the 'sliding joint' by the development of a ventral lamina on the skull roof to form a fossa and a distinctively shaped articular lamina on the trunk shield. Similarly, the 'reverse ginglymoid' presumably can be evolved from the sliding joint by the development of the dorsal articular lamina on the trunk shield. To better illustrate the transitions between various types of neck joint, we here propose the following characters relating to dermal neck joint for subsequent phylogenetic analyses.

(1) Presence or absence of a dermal head–trunk boundary. The dermal head–trunk boundary is absent in jawless stem gnathostomes and present in all jawed vertebrates with exceptions such as the arthrodire *Synauchenia* [55].

(2) Presence or absence of a dermal neck joint. The dermal neck joint is present in most placoderm subgroups except in rhenanids, and in many lower osteichthyans. It is absent in acanthodians and some nested osteichthyan groups such as tetrapods; non-applicable in conventionally defined chondrichthyans which completely lack a macromeric dermal skeleton.

(3) Presence or absence of a dorsal lamina along the anterodorsal margin of trunk shield. The dorsal lamina is present in antiarchs, *Qilinyu* and *Silurolepis*; absent in all other taxa. Together with the ventral lamina shared in most other placoderms, it defines the slot-shaped articular fossa for the 'reverse ginglymoid' on the trunk shield.

(4) Presence or absence of a distinct process/lamina to form a cranial fossa receiving the articular lamina. It is present in the *Qilinyu* with the dual articulation, and those placoderms in which the neck joints are categorized as 'ginglymoid' type; absent in antiarchs and those taxa with sliding neck joints.

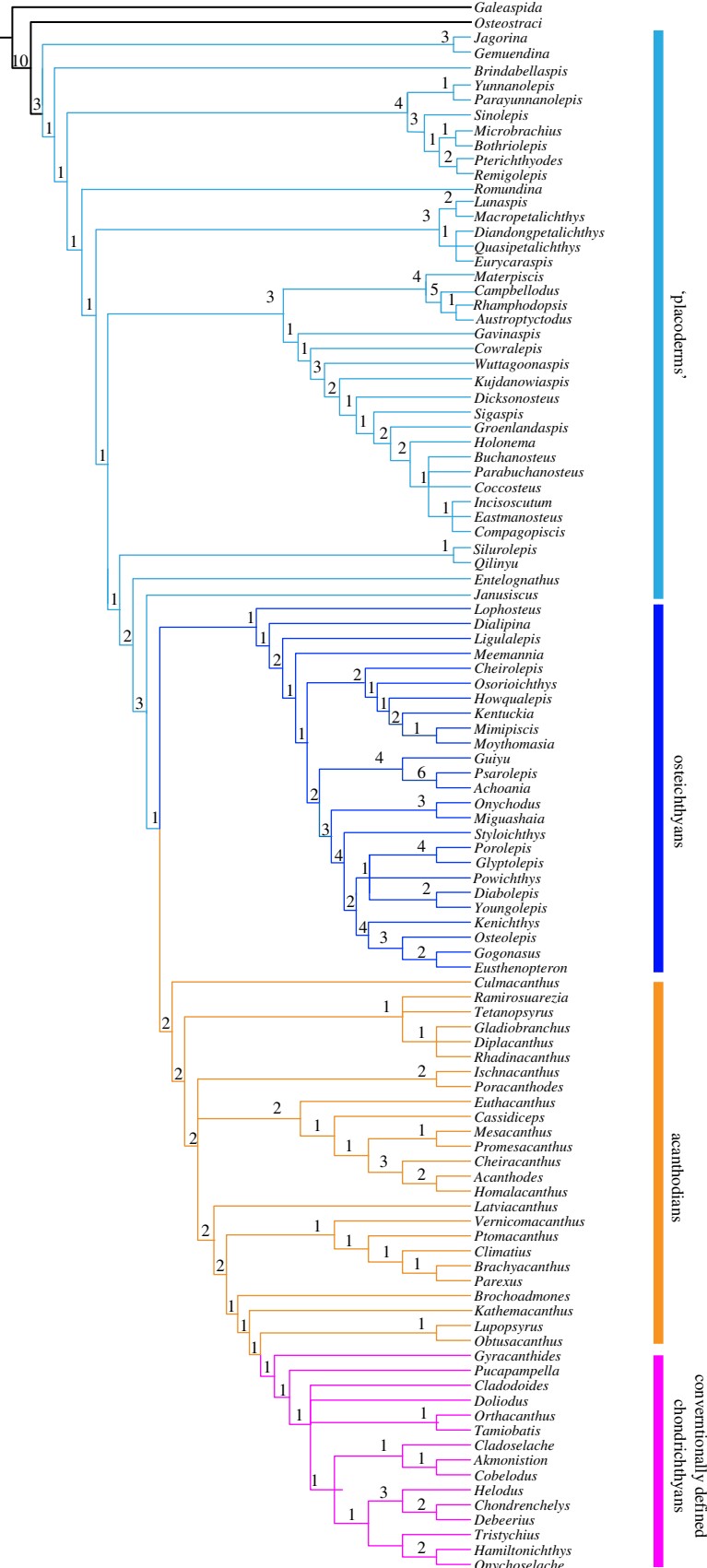

**Figure 5.** Strict consensus tree of 22 most parsimonious trees showing the phylogenetic position of *Silurolepis*. It is based on a character dataset with 379 characters and 105 taxa (tree length = 1045, consistency index = 0.3809). Numbers above branches denote Bremer decay indices.

(5) Ventral articular lamina on the trunk shield develops into flange or condyle distinct from the rest of the anterior margin of ADL. The well-defined flange or condyle is present in *Qilinyu*, *Eurycaraspis*, ptyctodonts and many arthrodires and absent in *Entelognathus* and basal arthrodire *Bryantolepis*.

(6) Presence or absence of a rotatory contact between articular lamina or condyle and the articular fossa. It is only present in arthrodires above phlyctaenids (including brachythoracids).

(7) The posterior margin of the skull roof significantly overlapped by the anterior-most median plate of the trunk shield or not. The first median plate in the trunk shield was variously named the extrascapular, the posteronuchal and the first MD plate in different placoderm groups. The exact homology between these bones is not well established. However, it is possible to distinguish two types of median contact relationship between the skull roof and trunk shield: (i) no significant overlap, such as nested brachythoracid arthrodires with large nuchal gap; (ii) skull roof strongly overlapped by the trunk shield, normally with a posteriorly convex nuchal plate but not necessarily so, as in antiarchs, *Qilinyu*, *Silurolepis* and some petalichthyids and ptyctodonts. In petalichthyids such as *Eurycaraspis*, the overlapped area is well defined laterally from the rest of the skull roof and is named the posterior descending lamina.

(8) Presence or absence of a lateral ridge or process (mesial articular process in [48]) defines the mesial boundary of the visceral contact face for the articular lamina on the paranuchal plate. The lateral ridge is present in petalichthyids and some ptyctodontids and absent in other placoderms.

(9) Presence or absence of an obstanic groove on the anterolateral margin of the trunk shield, formed by an extended anterior lamina curling laterally. It is present only in *Qilinyu* and *Silurolepis*.

## 4.3. Phylogenetic implications

Added into the data matrix from Zhu *et al.* [8] (electronic supplementary material, Silurolepis+matrix+final.nex), the above nine characters have the impact of uniting ptyctodontids and arthrodires together. Otherwise, the general topology of early gnathostomes remains stable (figure 5). *Silurolepis* is resolved to be the sister group of *Qilinyu* and the two form a clade immediately below *Entelognathus*. Notably, the distribution of dermal neck joint characters adds multiple evolutionary steps under the current phylogenetic framework, in which antiarchs are placed at the very basal section of jawed vertebrates, and *Qilinyu* plus *Silurolepis* are placed much more crownward, separated by a long array of paraphyletic placoderm subgroups (figure 4). This scenario implies that the 'reverse ginglymoid', or the presence of a dorsal lamina along the anterodorsal margin of trunk shield to form a slot-shaped articular fossa, evolved twice in antiarchs and maxillate placoderms, respectively. Also, the simple sliding type of neck joint in *Entelognathus* is suggested to be a reversal from the complicated dual articulation in *Qilinyu* and *Silurolepis*. Future synthesis on the character evolution in early jawed vertebrates is expected to shed further light on this conflicting evidence. One of the key tests is whether, in future analyses, the shared characters between *Entelognathus*, *Qilinyu* and crown-group gnathostomes are resolved as primitive or derived in jawed vertebrates. In the former case, these characters are expected to be also shared in some other placoderm subgroups, and the great disparity between the trunk shield morphology of *Qilinyu* and *Entelognathus* might in fact imply their distant systematic positions, distributed separately in the lineages from the initial radiation of jawed vertebrates.

Data accessibility. The CT data that support the findings of this study, as well as the three-dimensional surface files of IVPP V20732, are available in the IVPP Digital data repository ADMorph (Archives of Digital Morphology, http://dx.doi.org/10.12112/F.14).

Authors' contributions. M.Z. initiated the study, Y-a.Z. and M.Z. carried out phylogenetic analyses, Y-a.Z. and J.L. contributed new micro-CT data, segmentation and renderings, all authors contributed to writing the paper.

Competing interests. The authors declare there are no competing interests.

Funding. This study was supported by Strategic Priority Research Program of the Chinese Academy of Sciences (grant nos. XDA19050102, XDB26000000); National Natural Science Foundation of China (grant no. 41530102); Key Research Program of Frontier Sciences, CAS (grant no. QYZDJ-SSW-DQC002); National Natural Science Foundation of China (41872023) to J.L.; Swedish Research Council (grant no. 2014-4102) and a Wallenberg Scholarship from the Knut and Alice Wallenberg Foundation to Y-a.Z.

Acknowledgements. We thank Cui-hua Xiong for specimen preparation, Ye-Mao Hou for CT scanning, Lian-tao Jia for photography.

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
