## [Reviewer comments · Royal Society Open Science]

Review History

RSOS-191181.R0 (Original submission)

Review form: Reviewer 1

Is the manuscript scientifically sound in its present form?

Yes

Are the interpretations and conclusions justified by the results?

Yes

Is the language acceptable?

Yes

Do you have any ethical concerns with this paper?

No

Have you any concerns about statistical analyses in this paper?

No

Recommendation?

Accept with minor revision (please list in comments)

Comments to the Author(s)

A very interesting paper, and in particular the description of the neck joint in the 'maxillate placoderms'. This is a neglected area of placoderm anatomy, and an important one. I have minor editorial comments on the attached annotated manuscript, along with comments on Figures 3-5 (Appendix A).

Review form: Reviewer 2

Is the manuscript scientifically sound in its present form?

Yes

Are the interpretations and conclusions justified by the results?

Yes

Is the language acceptable?

Yes

Do you have any ethical concerns with this paper?

No

Have you any concerns about statistical analyses in this paper?

No

Recommendation?

Accept as is

Comments to the Author(s)

This manuscript redescribes, with CT scan images and 3D reconstructions, the Silurian 'placoderm' *Silurolepis*, long considered as the earliest antiarch. The authors conclude that *Silurolepis* is in fact a close relative of the other Silurian 'placoderm' *Qilinyu*, which is a 'maxillate placoderm', the two taxa forming a clade that diverged just before the 'maxillate placoderm' *Entelognathus* and the osteichthyans. 'Placoderms' have long been regarded as characterized by the presence of a dermal neck joint at the level of the limit between the skull roof and the trunk armour, and which coincides with an endoskeletal neck joint between the vertebral column and the braincase. The origin of this dermal neck joint has raised much debates among placoderm specialists, notably because of the different condition in antiarchs and arthrodires, with a condyle on the skull roof and an articular fossa on the trunk armour in the former, and the reverse in the latter. Thanks to the new data provided by *Silurolepis* and *Qilinyu*, the authors propose a new scenario of this evolution from a simple sliding dermal joint to the more complex condition in advanced 'placoderms', which foreshadows the condition in osteichthyans. The phylogenetic pattern of the 'placoderms' and their relationships to crown-group gnathostomes has been considerably reworked during the last 10 years with the groundbreaking discovery of the Silurian 'placoderms' and osteichthyans from China. This paper is an example of the wealth of new information provided by this new material of the earliest known jawed vertebrates. The new consideration of the basal segment of jawed vertebrate phylogeny will have to proceed stepwise, by re-visiting old combinations of characters, as it is done here. Now, the big enigma remains the transition between the condition in armored jawless vertebrates (e.g., osteostracans or galeaspids) and jawed vertebrates, that is, how the first neck joint arose. It is possible that it has to do with changes in the regulation of the organization of the paraxial mesoderm, or the inductions between the mesoderm and the neural crest derived skeletogenic

tissues of the dermoskeleton, but it remains uncertain whether this will be documented by fossils. We eagerly await the discovery of a jawless vertebrate with an independent shoulder girdle. As a whole, the manuscript is well written and informative, thanks to excellent illustrations.

Minor remarks:

- L. 25: conflicts with the parsimonious
- L. 56: chronologically closer (not chronically)
- L. 60: evidence always invariable in English
- L.64: evidence always invariable in English
- L.140: arching along the median line,
- L.141: ...an angle of approximately...
- L. 159: triangular in shape...
- L.219...neck adapting to the terrestrial...
- L.343: below Entelognathus (remove 'the')

Review form: Reviewer 3

Is the manuscript scientifically sound in its present form?

Yes

Are the interpretations and conclusions justified by the results?

Yes

Is the language acceptable?

Yes

Do you have any ethical concerns with this paper?

No

Have you any concerns about statistical analyses in this paper?

No

Recommendation?

Accept as is

Comments to the Author(s)

A most interesting paper with sound conclusions, excellent illustrations, and highlighting some very important early placoderm fossils from China. As I can see nothing to attend to before publication.

Decision letter (RSOS-191181.R0)

13-Aug-2019

Dear Dr Zhu

On behalf of the Editors, I am pleased to inform you that your Manuscript RSOS-191181 entitled "Reappraisal of the Silurian placoderm *Silurolepis* and insights into the dermal neck joint

evolution" has been accepted for publication in Royal Society Open Science subject to minor revision in accordance with the referee suggestions. Please find the referees' comments at the end of this email.

The reviewers and handling editors have recommended publication, but also suggest some minor revisions to your manuscript. Therefore, I invite you to respond to the comments and revise your manuscript.

- Ethics statement

- Data accessibility

If you wish to submit your supporting data or code to Dryad (<http://datadryad.org/>), or modify your current submission to dryad, please use the following link:
<http://datadryad.org/submit?journalID=RSOS&manu=RSOS-191181>

- Competing interests

- Authors' contributions

- Acknowledgements

- Funding statement

Because the schedule for publication is very tight, it is a condition of publication that you submit the revised version of your manuscript before 22-Aug-2019. Please note that the revision deadline will expire at 00.00am on this date. If you do not think you will be able to meet this date please let me know immediately.

Supplementary files will be published alongside the paper on the journal website and posted on the online figshare repository (<https://rs.figshare.com/>). The heading and legend provided for each supplementary file during the submission process will be used to create the figshare page,

so please ensure these are accurate and informative so that your files can be found in searches. Files on figshare will be made available approximately one week before the accompanying article so that the supplementary material can be attributed a unique DOI.

on behalf of Dr Julia Brenda Desojo (Associate Editor) and Kevin Padian (Subject Editor)
openscience@royalsociety.org

Reviewer comments to Author:
Reviewer: 1

Comments to the Author(s)

A very interesting paper, and in particular the description of the neck joint in the 'maxillate placoderms'. This is a neglected area of placoderm anatomy, and an important one. I have minor editorial comments on the attached annotated manuscript, along with comments on Figures 3-5.

Reviewer: 2

Comments to the Author(s)

This manuscript redescrines, with CT scan images and 3D reconstructions, the Silurian 'placoderm' *Silurolepis*, long considered as the earliest antiarch. The authors conclude that *Silurolepis* is in fact a close relative of the other Silurian 'placoderm' *Qilinyu*, which is a 'maxillate placoderm', the two taxa forming a clade that diverged just before the 'maxillate placoderm' *Entelognathus* and the osteichthyans. 'Placoderms' have long been regarded as characterized by the presence of a dermal neck joint at the level of the limit between the skull roof and the trunk armour, and which coincides with an endoskeletal neck joint between the vertebral column and the braincase. The origin of this dermal neck joint has raised much debates among placoderm specialists, notably because of the different condition in antiarchs and arthrodires, with a condyle on the skull roof and an articular fossa on the trunk armour in the former, and the reverse in the latter. Thanks to the new data provided by *Silurolepis* and *Qilinyu*, the authors

propose a new scenario of this evolution from a simple sliding dermal joint to the more complex condition in advanced 'placoderms', which foreshadows the condition in osteichthyans. The phylogenetic pattern of the 'placoderms' and their relationships to crown-group gnathostomes has been considerably reworked during the last 10 years with the groundbreaking discovery of the Silurian 'placoderms' and osteichthyans from China. This paper is an example of the wealth of new information provided by this new material of the earliest known jawed vertebrates. The new consideration of the basal segment of jawed vertebrate phylogeny will have to proceed stepwise, by re-visiting old combinations of characters, as it is done here. Now, the big enigma remains the transition between the condition in armored jawless vertebrates (e.g., osteostracans or galeaspids) and jawed vertebrates, that is, how the first neck joint arose. It is possible that it has to do with changes in the regulation of the organization of the paraxial mesoderm, or the inductions between the mesoderm and the neural crest derived skeletogenic tissues of the dermoskeleton, but it remains uncertain whether this will be documented by fossils. We eagerly await the discovery of a jawless vertebrate with an independent shoulder girdle. As a whole, the manuscript is well written and informative, thanks to excellent illustrations.

Minor remarks:

- L. 25: conflicts with the parsimonious
- L. 56: chronologically closer (not chronically)
- L. 60: evidence always invariable in English
- L.64: evidence always invariable in English
- L.140: arching along the median line,
- L.141: ...an angle of approximately...
- L. 159: triangular in shape...
- L.219...neck adapting to the terrestrial...
- L.343: below Entelognathus (remove 'the')

Reviewer: 3

Comments to the Author(s)

A most interesting paper with sound conclusions, excellent illustrations, and highlighting some very important early placoderm fossils from China. As I can see nothing to attend to before publication.

Author's Response to Decision Letter for (RSOS-191181.R0)

See Appendix B.

Decision letter (RSOS-191181.R1)

20-Aug-2019

Dear Dr Zhu,

I am pleased to inform you that your manuscript entitled "Reappraisal of the Silurian placoderm *Silurolepis* and insights into the dermal neck joint evolution" is now accepted for publication in Royal Society Open Science.

on behalf of Dr Julia Brenda Desojo (Associate Editor) and Kevin Padian (Subject Editor)
openscience@royalsociety.org

Follow Royal Society Publishing on Twitter: [@RSocPublishing](https://twitter.com/RSocPublishing)
Follow Royal Society Publishing on Facebook:
<https://www.facebook.com/RoyalSocietyPublishing.FanPage/>
Read Royal Society Publishing's blog: <https://blogs.royalsociety.org/publishing/>

Appendix A**ROYAL SOCIETY
OPEN SCIENCE****Reappraisal of the Silurian placoderm *Silurolepis* and
insights into the dermal neck joint evolution**

Journal:	Royal Society Open Science
Manuscript ID	RSOS-191181
Article Type:	Research
Date Submitted by the Author:	06-Jul-2019
Complete List of Authors:	Zhu, Youan; Institute of Vertebrate Paleontology and Paleoanthropology Chinese Academy of Sciences; Uppsala University Lu, Jing; Institute of Vertebrate Paleontology and Paleoanthropology Chinese Academy of Sciences Zhu, Min; Institute of Vertebrate Paleontology and Paleoanthropology Chinese Academy of Sciences
Subject:	evolution < BIOLOGY, palaeontology < BIOLOGY
Keywords:	placoderms, gnathostomes, neck joint, Silurian, Silurolepis
Subject Category:	Biology (whole organism)

Author-supplied statements

Relevant information will appear here if provided.

Ethics

Does your article include research that required ethical approval or permits?:

This article does not present research with ethical considerations

Statement (if applicable):

CUST_IF_YES_ETHICS :No data available.

Data

It is a condition of publication that data, code and materials supporting your paper are made publicly available. Does your paper present new data?:

Yes

Statement (if applicable):

The CT data that support the findings of this study, as well as the 3D surface files of IVPP V20732, are available in the IVPP Digital data repository ADMorph (Archives of Digital Morphology, <http://dx.doi.org/10.12112/F.14>)

Conflict of interest

I/We declare we have no competing interests

Statement (if applicable):

CUST_STATE_CONFLICT :No data available.

Authors' contributions

This paper has multiple authors and our individual contributions were as below

Statement (if applicable):

M.Z. initiated the study, Y.A.Z., M.Z. carried out phylogenetic analyses, Y.A.Z., J. L. contributed new micro-CT data, segmentation and renderings, all authors contributed to writing the paper

[revised manuscript text omitted]

5.3 Phylogenetic implications

Added into the data matrix from Zhu *et al.* [8], the above nine characters have the impact of uniting ptyctodontids and arthrodire together. Otherwise the general topology of early gnathostomes remains stable (figure 5). *Silurolepis* is resolved to be the sister group of *Qilinyu* and the two form a clade immediately below the *Entelognathus*. Notably, the distribution of dermal neck joint characters adds multiple evolutionary steps under the current phylogenetic framework, in which antiarchs are placed at the very basal section of jawed vertebrates, and *Qilinyu* plus *Silurolepis* are placed much higher, separated by a long array of paraclitic placoderm subgroups (figure 4). This scenario implies that the “reverse ginglymoid”, or at least the condition on the trunk shield in this type of dermal neck joint, evolved twice in antiarchs and maxillate placoderms respectively. Also, the simple sliding type of neck joint in *Entelognathus* is suggested to be reversely evolved from the complicated dual articulation in *Qilinyu* and *Silurolepis*. Future synthesis on the character evolution in early jawed vertebrates is expected to shed further light on this conflict of evidences. One of the key tests is whether the shared characters between *Entelognathus*, *Qilinyu*, and crown-group gnathostomes are primitive or derived in jawed vertebrates. In the former case, these characters are expected to be also shared in some other placoderm subgroups, and the great disparity between the trunk shield morphology of *Qilinyu* and *Entelognathus* make much more sense.

Acknowledgments

We thank Cui-hua Xiong for specimen preparation, Ye-Mao Hou for CT scanning, Lian-tao Jia for photography.

Funding Statement

Strategic Priority Research Program of the Chinese Academy of Sciences (XDA19050102, XDB26000000); National Natural Science Foundation of China (41530102); Key Research Program of Frontier Sciences, CAS (Grant No. QYZDJ-SSW-

362 DQC002); National Natural Science Foundation of China (41872023) to J.L.; Swedish Research Council grant 2014-4102
 and a Wallenberg Scholarship from the Knut and Alice Wallenberg Foundation to Y.A.Z.

Data Accessibility

The CT data that support the findings of this study, as well as the 3D surface files of IVPP V20732, are available in the
 IVPP Digital data repository ADMorph (Archives of Digital Morphology, <http://dx.doi.org/10.12112/F.14>).

Competing Interests

The authors declare there are no competing interests.

Authors' Contributions

373 M.Z. initiated the study, Y.A.Z., M.Z. carried out phylogenetic analyses, Y.A.Z., J. L. contributed new micro-CT data,
 segmentation and renderings, all authors contributed to writing the paper.

References

- 1. Burrow, C.J. & Sues, H.-D. 2011 A partial articulated acanthodian from the Silurian of New Brunswick, Canada. *Can. J. Earth Sci.* **48**, 1329–1341. (doi:10.1139/e11-023).
- 2. Janvier, P. 1996 *Early Vertebrates*. Oxford, Clarendon Press; 393 p.
- 3. Brazeau, M.D. & Friedman, M. 2015 The origin and early phylogenetic history of jawed vertebrates. *Nature* **520**, 490–497. (doi:10.1038/nature14438).
- 4. Wang, S.-T. 1993 *Vertebrate biostratigraphy of the Middle Palaeozoic of China*. In *Palaeozoic Vertebrate Biostratigraphy and Biogeography* (ed. J.A. Long), pp. 252–276. London, Belhaven Press.
- 5. Zhang, G.-R., Wang, S.-T., Wang, J.-Q., Wang, N.-Z. & Zhu, M. 2010 A basal antiarch (placoderm fish) from the Silurian of Qujing, Yunnan, China. *Palaeoworld* **19**, 129–135. (doi:10.1016/j.palwor.2009.11.006).
- 6. Zhu, M., Yu, X.-B., Ahlberg, P.E., Choo, B., Lu, J., Qiao, T., Qu, Q.-M., Zhao, W.-J., Jia, L.-J., Blom, H., et al. 2013 A Silurian placoderm with osteichthyan-like marginal jaw bones. *Nature* **502**, 188–193. (doi:10.1038/nature12617).
- 7. Zhu, M., Zhao, W.-J., Jia, L.-T., Lu, J., Qiao, T. & Qu, Q.-M. 2009 The oldest articulated osteichthyan reveals mosaic gnathostome characters. *Nature* **458**, 469–474.
- 8. Zhu, M., Ahlberg, P.E., Pan, Z.-H., Zhu, Y.-A., Qiao, T., Zhao, W.-J., Jia, L.-T. & Lu, J. 2016 A Silurian maxillate placoderm illuminates jaw evolution. *Science* **354**, 334–336. (doi:10.1126/science.aah3764).
- 9. Zhao, W.-J. & Zhu, M. 2015 A review of Silurian fishes from Yunnan, China and related biostratigraphy. *Palaeoworld* **24**, 243–250. (doi:10.1016/j.palwor.2015.02.004).
- 10. Zhao, W.-J. & Zhu, M. 2014 A review of the Silurian fishes from China, with comments on the correlation of fish-bearing strata. *Earth Sci. Front.* **21**, 185–202.
- 11. Zhu, Y.-A., Ahlberg, P. & Zhu, M. 2014 The Evolution of Vertebrate Dermal Jaw Bones in the Light of Maxillate Placoderms. In *Evolution and Development of Fishes* (eds. Z. Johanson, C. Underwood & M. Richter), pp. 71–86. London, Cambridge University Press.
- 12. Young, G.C. 2010 Placoderms (armored fish): dominant vertebrates of the Devonian period. *Annu. Rev. Earth Planet. Sci.* **38**, 523–550. (doi:10.1146/annurev-earth-040809-152507).
- 13. Brazeau, M.D., Friedman, M., Jerve, A. & Atwood, R.C. 2017 A three-dimensional placoderm (stem-group gnathostome) pharyngeal skeleton and its implications for primitive gnathostome pharyngeal architecture. *J. Morphol.* **278**, 1220–1228. (doi:10.1002/jmor.20706).
- 14. Goujet, D.F. 2001 Placoderms and basal gnathostome apomorphies. In *Major Events in Early Vertebrate Evolution: Palaeontology, Phylogeny, Genetics and Development* (ed. P.E. Ahlberg), pp. 209–222. London, Taylor & Francis.
- 15. Anderson, P.S.L., Friedman, M., Brazeau, M.D. & Rayfield, E.J. 2011 Initial radiation of jaws demonstrated stability despite faunal and environmental change. *Nature* **476**, 206–209. (doi:10.1038/nature10207).
- 16. Carr, R.K. 1995 Placoderm diversity and evolution. *Bull. Mus. Natl Hist. Nat. Ser. 4., Section C* **17**, 85–125.
- 17. Rücklin, M., Donoghue, P.C.J., Johanson, Z., Trinajstić, K., Marone, F. & Stampanoni, M. 2012 Development of teeth and jaws in the earliest jawed vertebrates. *Nature* **491**, 748–751. (doi:10.1038/nature11555).
- 18. Brazeau, M.D. & Friedman, M. 2014 The characters of Palaeozoic jawed vertebrates. *Zool. J. Linn. Soc.* **170**, 779–821. (doi:10.1111/zoj.12111).
- 19. Smith, M.M. & Johanson, Z. 2015 Origin of the vertebrate dentition: teeth transform jaws into a biting force. In *Great transformations in vertebrate evolution* (eds. K.P. Dial, N.H. Shubin & E.L. Brainerd), pp. 1–29. Chicago, The University of Chicago Press.
- 20. Zhu, M. & Wang, J.-Q. 2000 Silurian vertebrate assemblages of China. *Cour. Forsch.-Inst. Senckenberg* **223**, 161–168.

- 21. Zhang, G.-R. & Young, G.C. 1992 A head-trunk interface in tetrapod
 new antiarch (placoderm fish) from the vertebrates. *Elife* **5**, e09972.
 Early Devonian of South China. (doi:10.7554/eLife.09972).
 *Alcheringa* **16**, 219–240. 549
- 22. Zhu, M. 1996 The phylogeny of the 550
 Antiarcha (Placodermi, Pisces), with the 551
 description of Early Devonian antiarchs 552
 from Qujing, Yunnan, China. *Bull. Mus.* 553
 *Natl. Hist. Nat. C* **18**, 233–347. 554
- 23. Young, G.C. 1990 New antiarchs 555
 (Devonian placoderm fishes) from 556
 Queensland, with comments on 557
 placoderm phylogeny and biogeography 558
 *Mem. Queensl. Mus.* **28**, 35–50. 559
- 24. Goujet, D.F. 1973 *Sigaspis*, un nouveau 560
 arthrodire du Dévonien inférieur du 561
 Spitsberg. *Palaeontogr. Abt. A* **143**, 73 562
 88. 563
- 25. Zhu, M. & Schultze, H.-P. 2001 564
 Interrelationships of basal osteichthyans 565
 In *Major Events in Early Vertebrate 566*
 Evolution: Palaeontology, Phylogeny, 567
 Genetics and Development (ed. P. 568
 Ahlberg), pp. 289–314. London, Taylor 569
 Francis. 570
- 26. Miles, R.S. & Young, G.C. 1977 571
 Placoderm interrelationships 572
 reconsidered in the light of new 573
 ptyctodontids from Gogo, Western 574
 Australia. In *Problems in Vertebrate 575*
 Evolution (eds. S.M. Andrews, R.S. 576
 Miles & A.D. Walker), pp. 123–198. 577
 London, Academic Press. 578
- 27. Miles, R.S. & Westoll, T.S. 1968 The 579
 placoderm fish *Cocosteus cuspidatus* 580
 Miller ex Agassiz from the Middle Old 581
 Red Sandstone of Scotland. Part I. 582
 descriptive morphology. *Trans. R. Soc.* 583
 *Edinb. (Earth Sci.)* **67**, 373–476. 584
- 28. Young, G.C. 2009 New arthrodire 585
 (Family Williamsaspididae) from Wee 586
 Jasper, New South Wales (Early 587
 Devonian), with comments on 588
 placoderm morphology and 589
 palaeoecology. *Acta. Zool.* **90**, 69–82. 590
 (doi:10.1111/j.1463-6395.2008.00366.x) 591
- 29. Lours-Calet, C., Alvares, L.E., El- 592
 Hanfy, A.S., Gandesha, S., Walters, 593
 E.H., Sobreira, D.R., Wotton, K.R., 594
 Jorge, E.C., Lawson, J.A., Kelsey Lewis, 595
 537 A., et al. 2014 Evolutionarily conserved 596
 morphogenetic movements at the 597
 vertebrate head-trunk interface 598
 coordinate the transport and assembly 599
 of hypopharyngeal structures. *Dev. Biol.* 600
 **390**, 231–246. 601
 (doi:10.1016/j.ydbio.2014.03.003). 602
- 30. Sefton, E.M., Bhullar, B.A., Mohammed 603
 Z. & Hanken, J. 2016 Evolution of the 604
- Wilson & R. Cloutier), pp. 109–126.
 München, Verlag Dr. Friedrich Pfeil.
- 43. Dupret, V. 2010 Revision of the genus
 *Kujdanowiaspis* Stensiö, 1942
 (Placodermi, Arthrodira, "Actinolepida")
 from the Lower Devonian of Podolia
 (Ukraine). *Geodiversitas* **32**, 5–63.
 (doi:10.5252/g2010n1a1).
- 44. Young, G.C. 1980 A new Early
 Devonian placoderm from New South
 Wales, Australia, with a discussion of
 placoderm phylogeny. *Palaeontogr. Abt.*
 **A 167**, 10–76.
- 45. Wang, Y. & Zhu, M. 2018
 Redescription of *Phymolepis*
 *cuijingshanensis* (Antiarcha:
 Yunnanolepididae) using high-resolution
 computed tomography and new insights
 into anatomical details of the
 endocranium in antiarchs. *PeerJ* **6**,
 e4808.
- 46. Goujet, D.F. 1984 Les poissons
 placodermes du Spitsberg. Arthrodirés
 Dolicho thoraci de la Formation de Wood
 Bay (Dévonien inférieur). Paris, Editions
 Centre National Recherche Scientifique,
 Cahiers de Paléontologie; 284 p.
- 47. Liu, Y.-H. 1991 On a new
 petalichthyid, *Eurycaraspis incilis* gen. et
 sp. nov., from the Middle Devonian of
 Zhanyi, Yunnan. In *Early Vertebrates*
 and Related Problems of Evolutionary
 Biology (eds. M.-M. Chang, Y.-H. Liu &
 G.-R. Zhang), pp. 139–177. Beijing,
 Science Press.
- 48. Zhu, M. 1991 New information on
 *Diandongpetalichthys* (Placodermi:
 Petalichthyida). In *Early Vertebrates and*
 *Related Problems of Evolutionary*
 *Biology* (eds. M.-M. Chang, Y.-H. Liu &
 G.-R. Zhang), pp. 179–194.
- 49. Trinajstić, K., Long, J.A., Johanson, Z.,
 Young, G. & Senden, T. 2012 New
 morphological information on the
 ptyctodontid fishes (Placodermi,
 Ptyctodontida) from Western Australia.
 *J. Vert. Paleont.* **32**, 757–780.
 (doi:10.1080/02724634.2012.661379).
- 50. Trinajstić, K. & Long, J.A. 2009 A new
 genus and species of Ptyctodont
 (Placodermi) from the Late Devonian
 Gneudna Formation, Western Australia,
 and an analysis of Ptyctodont
 phylogeny. *Geol. Mag.* **146**, 743–760.
 (doi:10.1017/s001675680900644x).
- 51. Long, J.A. 1997 Ptyctodontid fishes
 (Vertebrata, Placodermi) from the Late
 Devonian Gogo Formation, Western
 Australia, with a revision of the

- European genus *Ctenurella* Ørvig, 1966 672 (Arthrodira: Heterostiidae) from the 680 the organization of the head in the
 665 *Geodiversitas* **19**, 515–555. 673 Lower Devonian of China, and the 681 Dolichothoraci, Coccoosteomorphi and
 666 52. Stensiö, E.A. 1959 On the pectoral fin 674 interrelationships of Brachythoraci. *Zool.* 682 Pachyosteomorphi. Taxonomic
 and shoulder girdle of the arthrodiras. 675 *J. Linn. Soc.* **176**, 806–834. 683 appendix. *Kungl. Svenska. Vetenskap.*
 *Kungl. Svenska. Vetenskap. Hand.* **8**, 676 (doi:10.1111/zoj.12356). 684 *Hand.* **9**, 1–419.
 229. 677
 53. Zhu, Y.-A., Zhu, M. & Wang, J.-Q. 678 54. Stensiö, E.A. 1963 Anatomical studies
 2016 Redescription of *Yinostius major* 679 on the arthrodiran head. Part 1. Preface,
 geological and geographical distribution,

685

686 **Figure captions**

[revised manuscript text omitted]

Figure 1. *Silurolepis platydorsalis* (holotype, IVPP V11680.1)

151x126mm (300 x 300 DPI)

Figure 2. Internal mould of the trunk shield of *Silurolepis platydorsalis*

144x168mm (300 x 300 DPI)

Figure 3. Digital reconstruction and rendering of *Qilinyu rostrate* (IVPP V20732).

144x138mm (300 x 300 DPI)

Figure 4. Evolution of dermal neck joint in the earliest jawed vertebrates

162x109mm (300 x 300 DPI)

Figure 5. Strict consensus tree of 22 most parsimonious trees showing the phylogenetic position of *Silurolepis*

101x226mm (300 x 300 DPI)

Appendix B

Reviewer comments to Author:

Reviewer: 1

Comments to the Author(s)

A very interesting paper, and in particular the description of the neck joint in the 'maxillate placoderms'. This is a neglected area of placoderm anatomy, and an important one. I have minor editorial comments on the attached annotated manuscript, along with comments on Figures 3-5.

Reply : We have made the modifications according to the annotated manuscript. See point to point replies bellow.

Line 17: Trunkshield is one word?

Reply: "trunk shield" is normally two words, but sometimes also one word in some publications, we tend to keep it in two words.

Line 104: Just a question- can this be a valid species name if it is a nomen nudum?

Reply: *Silurolepis platydorsalis* was a nomen nudum in Wang 1993 and Zhu & Wang, 2000, as in both papers the name is present but there lacks a proper description. In Zhang et al., 2010 it is described and formally erected as a scientific name for the taxa.

Line 263: Yes, there has to be an articulating surface on the trunkshield or there wouldn't be a neck?

Reply: There has to be a contacting surface, but NOT necessarily have to be an articular lamina under our definition (ventral flange on trunk armour, dorsally overlapped by the skull roof), whereas theoretically there can be the opposite condition – the trunk armour dorsally overlaps skull roof and lacks the ventral articular lamina, as happened in some nested osteichthyans, but to date never in stem gnathostomes.

Line 297: I don't understand what you mean?

Reply: We have revised the sentence as follows:

"The traditionally defined categories of neck joint types in placoderms obviously represent a composite of characters, and the transitions between these types can be achieved by the stepwise development or reduction of independent structures."

Line 302: This should be with reference to a phylogeny?

Reply: We have revised the sentence as follows:

“To better illustrate the transitions between various types of neck joint, we here propose following characters relating to dermal neck joint for subsequent phylogenetic analyses.”

Line 322: does this repeat character 4) the ginglymoid articulation?

Reply: No. This is a contingent character to the previous characters, but does not repeat or overlap the character 4). The traditionally defined “ginglymoid articulation” is a composition of characters including the rotatory condition. However, we here redefined the characters to avoid the composition and to illustrate the stepwise transitions. In character 4) it is denoted that a distinct process/lamina to form a cranial fossa is present in those placoderms in which the neck joints are categorized as “ginglymoid” type, this is not to say that the “ginglymoid” type equals the present condition in character 4), and the condition does not necessarily include rotatory contact, e.g. in *Qilinyu*.

Line 346: I don't understand what you mean by this?

Reply: We have revised the sentence as follows:

This scenario implies that the “reverse ginglymoid”, or the presence of a dorsal lamina along the anterodorsal margin of trunk shield to form a slot-shaped articular fossa, evolved twice in antiarchs and maxillate placoderms respectively.

Line 353: I don't understand how this highlighted part of the sentence follows from the first?

Reply: We have revised the sentence as follows:

In the former case, these characters are expected to be also shared in some other placoderm subgroups, and the great disparity between the trunk shield morphology of *Qilinyu* and *Entelognathus* might in fact implies their distant systematic positions, distributed separately in the lineages from the initial radiation of jawed vertebrate.

Figure 3, 4:

We have modified the figures according to the reviewer's comment.

Figure 5:

We have modified the figure, including *Janusiscus* in “Placoderm”, following Zhu et al., 2016.

Reviewer 2:

Comments to the Author(s)

This manuscript redescribes, with CT scan images and 3D reconstructions, the Silurian ‘placoderm’ *Silurolepis*, long considered as the earliest antiarch. The authors conclude that *Silurolepis* is in fact a close relative of the other Silurian ‘placoderm’ *Qilinyu*, which is a ‘maxillate placoderm’, the two taxa forming a clade that diverged just before the ‘maxillate placoderm’ *Entelognathus* and the osteichthyans. ‘Placoderms’ have long been regarded as characterized by the presence of a dermal neck joint at the level of the limit between the

skull roof and the trunk armour, and which coincides with an endoskeletal neck joint between the vertebral column and the braincase. The origin of this dermal neck joint has raised much debates among placoderm specialists, notably because of the different condition in antiarchs and arthrodires, with a condyle on the skull roof and an articular fossa on the trunk armour in the former, and the reverse in the latter. Thanks to the new data provided by *Silurolepis* and *Qilinyu*, the authors propose a new scenario of this evolution from a simple sliding dermal joint to the more complex condition in advanced 'placoderms', which foreshadows the condition in osteichthyans.

The phylogenetic pattern of the 'placoderms' and their relationships to crown-group gnathostomes has been considerably reworked during the last 10 years with the groundbreaking discovery of the Silurian 'placoderms' and osteichthyans from China. This paper is an example of the wealth of new information provided by this new material of the earliest known jawed vertebrates. The new consideration of the basal segment of jawed vertebrate phylogeny will have to proceed stepwise, by re-visiting old combinations of characters, as it is done here. Now, the big enigma remains the transition between the condition in armored jawless vertebrates (e.g., osteostracans or galeaspids) and jawed vertebrates, that is, how the first neck joint arose. It is possible that it has to do with changes in the regulation of the organization of the paraxial mesoderm, or the inductions between the mesoderm and the neural crest derived skeletogenic tissues of the dermoskeleton, but it remains uncertain whether this will be documented by fossils. We eagerly await the discovery of a jawless vertebrate with an independent shoulder girdle.

As a whole, the manuscript is well written and informative, thanks to excellent illustrations.

Minor remarks:

L. 25: conflicts with the parsimonious

L. 56: chronologically closer (not chronically)

L. 60: evidence always invariable in English

L.64: evidence always invariable in English

L.140: arching along the median line,

L.141: ...an angle of approximately...

L. 159: triangular in shape...

L.219...neck adapting to the terrestrial...

L.343: below *Entelognathus* (remove 'the')

Reply: We have made the corrections according to the remarks of Referee 2.

Reviewer: 3

Comments to the Author(s)

A most interesting paper with sound conclusions, excellent illustrations, and highlighting some very important early placoderm fossils from China. As I can see nothing to attend to before publication.

Reply: Thanks!